# Dapagliflozin mitigates myocardial inflammation and metabolic stress in heart failure through STAT1 inhibition: Evidence from multi-omics analyses and experimental exploration

Lianqi He[1], Di Zhang[1], Shiwen Zhang[1], Zhichao Ren[1], Yihong Huangfu[1,2], Xiran Cheng[3], Zhanchun Song[1]*

1 Department of Cardiology, Central Hospital of Fushun, Fushun, Liaoning, China, 2 Graduate School, Jinzhou Medical University, Jinzhou, Liaoning, China, 3 Changsha Medical College, Changsha, Hunan, China

* szcclszccl@163.com

## Abstract

### Background

Heart failure (HF) is a global health challenge with high morbidity and mortality. While dapagliflozin (DAPA), a sodium–glucose cotransporter 2 inhibitor, has proven clinical benefits in HF, its molecular mechanisms remain unclear.

### Methods

We integrated bulk and single-cell transcriptomic analyses with experimental validation to investigate the role of STAT1 in HF and its modulation by DAPA. Bulk RNA sequencing data from the GSE57345 dataset were analyzed for differential expression, enrichment, and weighted gene co-expression network analysis (WGCNA) to identify hub regulators. Single-cell RNA sequencing data (GSE145154) included normal controls (n = 4), dilated cardiomyopathy (DCM, n = 8), ischemic cardiomyopathy infarct regions (ICM_MI, n = 6), and non-infarct regions (ICM_NMI, n = 6), and were processed with Seurat and Harmony for integration, clustering, myeloid subcluster profiling, and AUCell pathway scoring. For in vivo validation, a rat model of myocardial infarction–induced HF was established and divided into control, HF, and HF+DAPA groups (6 mg/kg/day for 4 weeks). Histological examination, Western blotting, ELISA, flow cytometry, and serum bile acid assays were conducted. For in vitro assays, STAT1-overexpressing H9C2 cardiomyocytes were generated by lentiviral transduction. Cell viability (CCK-8), STAT1 expression (qPCR and Western blot), and apoptosis (Annexin V/PI flow cytometry) were assessed with or without DAPA treatment.

**Data availability statement:** All relevant data are within the manuscript and its Supporting information files.

**Funding:** This study was supported by the talent Program of Fushun City of China (No. FSYC202107012). The funders had no role in study design, data collection and analysis, decision to publish, or preparation of the manuscript.

**Competing interests:** The authors have declared that no competing interests exist.

## Results

Transcriptomic analyses revealed widespread activation of bile acid, amino acid, and lipoic acid metabolic pathways in HF, coupled with immune remodeling dominated by increased M1 and reduced M2 macrophages. STAT1 emerged as a central hub gene linking metabolic stress and immune imbalance. Single-cell analysis confirmed aberrant STAT1 expression particularly in M1 and proliferating myeloid clusters. In vivo, DAPA suppressed myocardial STAT1 expression, alleviated inflammation, normalized macrophage polarization, and reduced cytokine and bile acid abnormalities. In vitro, DAPA rescued STAT1-overexpressing cardiomyocytes by restoring viability and reducing apoptosis.

## Conclusions

STAT1 acts as a pivotal mediator bridging metabolic disturbances and immune dysregulation in HF. DAPA alleviates HF by inhibiting STAT1 signaling, thereby restoring immunometabolic balance and protecting cardiac tissue. These findings provide mechanistic insight into the cardioprotective effects of DAPA and position STAT1 as a promising biomarker and potential therapeutic candidate for HF management.

---

## 1.  1Introduction

Heart failure (HF) is a complex clinical syndrome characterized by the inability of the heart to pump sufficient blood to meet the body's metabolic demands, resulting from structural or functional cardiac abnormalities that impair ventricular filling or ejection [1]. It represents a major global health burden, affecting approximately 64.3 million individuals worldwide, and continues to be associated with high morbidity and mortality [2]. Despite considerable progress in pharmacological therapies (such as β-blockers, angiotensin-converting enzyme inhibitors, and angiotensin receptor–neprilysin inhibitors) and interventional strategies (implantable cardioverter-defibrillators and cardiac resynchronization therapy), the prognosis of HF remains poor, with 1-year and 3-year mortality rates reported to be 15–30% and 30–50%, respectively [3,4].

The pathogenesis of HF involves multiple interrelated processes, including myocardial remodeling, oxidative stress, metabolic disturbance, inflammation, and apoptosis [5,6]. In particular, metabolic stress and immune dysregulation are now recognized as central drivers of disease progression. Patients with HF frequently exhibit profound alterations in myocardial energy metabolism, ranging from abnormal fatty acid oxidation to shifts in amino acid and ketone body utilization, accompanied by systemic metabolic changes in bile acid and glucose pathways [7]. Such metabolic reprogramming interacts closely with the immune microenvironment, which is characterized by enhanced pro-inflammatory responses and imbalance of macrophage polarization [8,9]. Together, metabolic and immunological stressors form a maladaptive network that accelerates cardiomyocyte injury, fibrosis, and functional decline.

Transcription factors have emerged as pivotal regulators in this context. Signal transducer and activator of transcription 1 (STAT1) is a key downstream effector of classical JAK–STAT signaling, broadly involved in inflammatory activation and apoptosis [10,11]. Increasing evidence suggests that STAT1 overactivation contributes to HF progression by enhancing ferroptosis, apoptosis, and inflammatory responses [12,13]. However, its precise role in coordinating metabolic stress with immune dysregulation in HF tissues, especially at the single-cell level, remains largely unexplored [14,15].

Sodium–glucose cotransporter 2 (SGLT2) inhibitors, such as dapagliflozin (DAPA), have attracted great interest owing to their ability to improve outcomes in HF, beyond their glucose-lowering properties [16,17]. Beyond systemic effects—including weight loss, blood pressure reduction, and metabolic regulation [18,19]—DAPA has been reported to exert direct cardioprotective effects by alleviating oxidative stress, inflammation, and myocardial remodeling [20–23]. Yet, the molecular mechanisms by which DAPA regulates intracellular signaling in HF remain incompletely understood. Notably, whether DAPA modulates the STAT1 pathway to mediate its cardioprotective effects has not been directly confirmed.

In the present study, we used a combination of bulk and single-cell transcriptome analyses, together with in vivo and in vitro experiments, to investigate the role of STAT1 in HF progression and its modulation by DAPA. We aimed to clarify how metabolic stress–related changes in the immune microenvironment are linked with STAT1 expression at the single-cell level, and to determine whether DAPA improves HF outcomes by suppressing STAT1 signaling. This integrated approach provides new mechanistic insights into HF pathogenesis and suggests STAT1 expression may be modulated during DAPA-mediated intervention.

## 2. Materials and methods

### 2.1 Acquisition and processing of bulk transcriptomic data

The myocardial transcriptomic dataset GSE57345 was obtained from the Gene Expression Omnibus (GEO) database. After evaluating data quality and performing normalization on the raw count matrix, differential expression analysis was conducted using DESeq2 (R package), with thresholds set at $P < 0.05$ and $|\log_2 \text{FoldChange}| > 0.5$ to identify differentially expressed genes (DEGs). Functional enrichment analyses, including GO and KEGG pathways, were performed with clusterProfiler to elucidate the biological processes and signaling pathways associated with the DEGs [24]. For immune infiltration profiling, CIBERSORT was applied to deconvolute bulk expression data and to estimate the relative abundance of 22 immune cell types. All analyses were performed in the R environment, with visualization using ggplot2. Statistical tests included two-sided t-tests or one-way ANOVA, with significance defined as $P < 0.05$.

### 2.2 Acquisition of single-Cell RNA sequencing data and sample construction

Single-cell RNA sequencing (scRNA-seq) data were obtained from the GSE145154 dataset. Based on the study design, four clinical groups were included: normal controls (Control, n = 4), dilated cardiomyopathy (DCM, n = 8), ischemic cardiomyopathy infarct regions (ICM_MI, n = 6), and non-infarct regions (ICM_NMI, n = 6). For each sample, 10x Genomics files were read and converted into individual Seurat objects, which were subsequently merged into a unified object while preserving sample metadata.

### 2.3 Quality control, normalization, and integration-clustering

For the publicly available human single-cell RNA sequencing data obtained from the GSE145154 dataset, quality control (QC) metrics were computed at the single-cell level, including mitochondrial transcript proportion, ribosomal gene proportion, and hemoglobin gene proportion. Violin and scatter plots were generated by sample before filtering. Cells meeting the following thresholds were retained: nFeature_RNA ≥ 200, nFeature_RNA ≤ 3000, nCount_RNA ≥ 500, nCount_RNA ≤ 10000, percent.mt ≤ 10%, and percent.hb ≤ 5%. Normalization was performed using LogNormalize, and highly variable genes (nfeatures = 3000) were identified using the vst method. ScaleData was then used to regress out

percent.mt. Principal component analysis (PCA) was performed on the variable genes, and the elbow plot guided the selection of significant PCs. To reduce batch effects across samples, Harmony integration was applied on PCA embeddings, using sample_id as the integration variable. Neighbor graph construction and clustering were performed on Harmony-corrected embeddings (resolution = 0.5), followed by UMAP/t-SNE visualization.

## 2.4 Cell-type annotation and composition analysis

Cluster-specific marker genes were identified, and canonical markers were applied for cell-type annotation. Key annotations included endothelial cells (EC: VWF, CLDN5), endocardial cells (EndoC: EMCN, EDN1), cardiomyocytes (CM: TNNT2, TNNI3), smooth muscle cells (SMC: ACTA2, MYH11), pericytes (RGS5, ABCC9), fibroblasts (FB: LUM, DCN), myeloid cells (LST1, AIF1, C1QC), NK cells (PRF1, KLRF1), T cells (CD3D/E, TRAC/TRBC), and B cells (MS4A1, CD79A/B) [25–27]. Visualization was performed using DotPlots and DimPlots. The proportion of each cell type per sample was calculated, averaged within each clinical group (Control, DCM, ICM_MI, and ICM_NMI), and visualized as bar plots with error bars (mean ± standard error (SE)).

## 2.5 Secondary clustering and subtype annotation of myeloid cells

Myeloid cells were subsetted from the merged object and reprocessed using the standard workflow. Subclusters were identified using FindAllMarkers, and annotated with specific marker panels: pro-inflammatory M1-like macrophages (IL1B, CCL3/4, NLRP3, S100A8/9), reparative M2-like macrophages (MRC1, CD163, APOE/C1, TREM2, GPNMB, LYVE1), homeostatic regulators (M0; IL1RN, KLF2/4, NR4A1, TXNIP), and proliferating cells (TOP2A, MKI67, BIRC5) [28–30]. Each cell was assigned a subtype, and distribution was visualized using UMAP and DotPlots. Subset composition per sample was summarized into group-level means ± SE and displayed using bar charts.

## 2.6 Pathway activity scoring (single-cell AUCell)

To address transcript dropout in STAT1 signaling at the single-cell level and to quantify pathway activity, AUCell was applied. Hallmark gene sets were retrieved from MSigDB, including HALLMARK_INTERFERON_GAMMA_RESPONSE, HALLMARK_INTERFERON_ALPHA_RESPONSE, and HALLMARK_IL6_JAK_STAT3_SIGNALING. Additionally, a curated STAT1 downstream target module was compiled from the literature (e.g., IRF1, CXCL10, ISG15, MX1, OAS1/2, DDX58, IFIT1/3, STAT1) [31–34]. Log-normalized expression matrices from both global and myeloid subsets were ranked using AUCell_buildRankings, and AUCell scores (AUC values; aucMaxRank ≈ 5% of total genes) were computed per cell using AUCell_calcAUC. Scores were incorporated into metadata, and distribution differences across clinical groups (Control, DCM, ICM_MI, and ICM_NMI) were analyzed at both the cell-type and myeloid-subtype levels. Results were visualized using violin and box plots.

## 2.7 Animal model and sampling

All animal experiments were approved by the Ethics Committee for the Care and Use of Laboratory Animals of Fushun Municipal Central Hospital (Approval No. 2022005) and followed relevant regulations and ARRIVE guidelines. Eighteen male Sprague–Dawley rats (12 weeks, ~ 160 g) were randomly allocated into a control group (CK, n = 6) and an experimental group (n = 12). Myocardial infarction–induced heart failure (HF) was created in the experimental group by ligation of the left anterior descending (LAD) coronary artery. Specifically, rats were anesthetized with isoflurane and fixed in the supine position after hair removal. Electrocardiogram (ECG) was continuously monitored throughout the procedure. After sterile preparation, a thoracotomy was performed along the left sternal border at the 3rd–4th intercostal space using tissue scissors to expose the heart. The left atrial appendage was gently lifted with sterile toothed forceps to expose the aortic root. A surgical suture was passed around the proximal segment of the left anterior descending coronary artery

near the superior margin of the heart, and three secure knots were tied to achieve complete ligation. The thoracic wall was then rapidly closed with sutures. Control rats underwent the same thoracotomy procedure without LAD ligation (sham operation). Post-operatively, all rats were placed in a warm recovery chamber and monitored until full consciousness was regained. Analgesics and antibiotics were administered as needed to minimize discomfort and prevent infection. Of the HF rats, six were randomly assigned to receive dapagliflozin (HF + DAPA). A pilot dosing study (3, 6, 9 mg/kg/day) identified 6 mg/kg/day as the most effective dose for reducing myocardial injury and inflammation (S1A Fig). Dapagliflozin was dissolved in 0.5% carboxymethylcellulose sodium (CMC-Na; HY-Y0703, MedChemExpress, USA) as the vehicle. Accordingly, from the second day after surgery, the HF + DAPA group was treated daily with oral gavage of dapagliflozin (HY-10450; MedChemExpress, USA, 6 mg/kg/day) dissolved in 0.5% CMC-Na for four weeks, while the HF group received an equivalent volume of 0.5% CMC-Na vehicle alone. The control group (CK) also received 0.5% CMC-Na vehicle. Before tissue sampling, deep anesthesia was administered via intraperitoneal injection of sodium pentobarbital (200 mg/kg) until respiratory arrest. Blood samples were collected from the orbital venous plexus, and hearts were harvested at sacrifice four weeks post-modeling.

## 2.8 Histological and protein expression analyses

For histopathology, myocardial tissues from the infarction area were fixed in 4% paraformaldehyde, dehydrated in graded sucrose, embedded in paraffin, and sectioned. Sections were dewaxed in xylene, rehydrated through ethanol gradients, and stained with hematoxylin and eosin (H&E). Morphological changes were observed under a light microscope.

For protein expression, heart tissues from the infarction area were minced and homogenized in cold lysis buffer containing protease and phosphatase inhibitors, incubated for 30 min on ice, and centrifuged (4 °C, 12,000 rpm, 20 min). Supernatants were quantified by BCA assay (Beyotime, China). Equal amounts of protein were mixed with loading buffer, boiled (10 min), denatured, separated by SDS–PAGE, and transferred to PVDF membranes. After blocking with 5% BSA for 1 h, membranes were incubated overnight at 4 °C with primary antibodies against STAT1 (1:1000 dilution; ET1612−22, Huabio, China) or GAPDH (1:5000 dilution; AF7021, Affinity, China) as the loading control. After washing three times with TBST, membranes were probed with goat anti-rabbit IgG secondary antibody (1:50,000 dilution; HA1001, Huabio, China) for 1 h at room temperature, and visualized using ECL substrate (AIWB-006; Affinity-life, China). Chemiluminescence images were captured and analyzed using ImageJ.

## 2.9 Immunological assays

Peripheral blood mononuclear cells (PBMCs) were isolated via Ficoll density gradient centrifugation (blood:PBS:Ficoll = 1:1:1, 400 g, 30 min). Collected PBMCs were washed with PBS, resuspended in 10% FBS medium, and enumerated before subsequent analysis. To evaluate macrophage polarization, cells were stained with APC-conjugated CD86 (M1 marker; BioLegend, USA) and PerCP/Cy5.5-conjugated CD163 (M2 marker; Biorbyt, UK) for 30 min at 4 °C in the dark, fixed with 4% paraformaldehyde, and analyzed on a FACS Calibur flow cytometer (BD, USA). Data were processed with FlowJo software to quantify CD86$^+$ and CD163$^+$ subsets.

Serum inflammatory cytokines were quantified using ELISA kits with absorbance recorded at 450 nm. TNF-α was measured using a rat TNF-α ELISA kit (CSB-E11987r-IS, Cusabio, China) with a detection range of 15.6–1000 pg/ml, sensitivity of 3.9 pg/ml, intra-assay CV < 8%, and inter-assay CV < 10%. IL-6 was measured using a rat IL-6 ELISA kit (CSB-E04640r, Cusabio, China) with a detection range of 0.312–20 pg/ml, sensitivity of 0.078 pg/ml, intra-assay CV < 8%, and inter-assay CV < 10%. Concentrations were calculated by four-parameter logistic regression. Total bile acid (TBA) levels in serum were measured using a colorimetric assay kit (E003-2–1; Nanjing Jiancheng Institute, China) with a detection range of 0–180 μmol/L, intra-assay CV ≤ 5.0%, and inter-assay CV ≤ 10.0%, according to the manufacturer's protocol. The assay is based on enzymatic cycling reactions: bile acids are oxidized by 3α-hydroxysteroid dehydrogenase (3α-HSD) in the presence of S-NAD to produce 3-keto steroids and S-NADH; subsequently, 3-keto steroids are reduced back

by S-NADH through diaphorase to regenerate $NAD^+$ and bile acids. This cycling amplification reaction produces a color change at 405 nm, with the absorbance proportional to the bile acid concentration in the sample [35].

## 2.10 Cell-based experiments

The H9C2 cardiomyocyte line was used for in vitro analyses. All cell culture experiments were conducted under standard normoxic conditions (37°C, 5% $CO_2$, atmospheric oxygen levels ~21% $O_2$) without oxygen deprivation or hypoxic treatment. To generate STAT1-overexpressing cells, cultures were infected with a lentiviral vector (HX-H-OELV-16263; Starfish Biology, China) in 24-well plates, incubated overnight, and subsequently expanded into 6-well plates. Stable cell lines were selected with G-418 (200 µg/mL) for 48 h, and resistant clones were used in downstream assays.

Cell viability was measured using the CCK-8 assay (LiXin Biology, China). Cells at 80–90% confluence were incubated with 10 µL of reagent for 1 h at 37 °C, and absorbance was read at 450 nm. For gene expression analysis, total RNA was extracted with TRIzol (LiXin Biology, China), reverse transcribed with a cDNA synthesis kit, and amplified by qPCR with SYBR Green Mix using β-actin as control. Primer sequences were STAT1-F (5′-TCTGGCCTTGGATTGACACC-3′) and STAT1-R (5′-TGAATGTGATGGCCCCTTCC-3′). Relative expression was calculated using the 2−ΔΔCT method.

Apoptosis was evaluated by Annexin V-FITC/PI double staining (Beyotime, China). Cells were resuspended at $2 \times 10^5$/ mL, incubated with binding buffer, Annexin V-FITC, and PI in the dark, and analyzed within 1 h on a BD LSRFortessa flow cytometer. Voltage and gating were set with blank and single-color controls, allowing quantification of early, late, and total apoptosis using FlowJo.

## 2.11 Statistical analysis and visualization

In the bioinformatics analysis section, for pathway scores (AUCell) within each cell_type or myeloid subtype, the Kruskal–Wallis test was applied to compare overall differences among the four groups (Control, DCM, ICM_MI, ICM_NMI). Pairwise comparisons were performed using Dunn's post hoc test, and multiple testing correction was conducted using the Benjamini–Hochberg method; significance levels were annotated in figures using symbols. Visualization was mainly performed with ggplot2, patchwork, and ggpubr; data wrangling was carried out using dplyr, tidyr, stringr, and purrr; dimensionality reduction and visualization were implemented via Seurat and SCP; batch correction was conducted using harmony. Gene naming and quality control regex matching were based on human gene symbols (mitochondrial MT-, ribosomal RPS/RPL, hemoglobin HB*).

In the experimental validation section, normality of data distribution was verified using the Shapiro-Wilk test prior to parametric analysis. All datasets met the assumption of normal distribution. Statistical significance between two groups was assessed with an unpaired two-tailed t test, while comparisons among multiple groups were performed using one-way analysis of variance (ANOVA) followed by Tukey's HSD test. For in vivo experiments, Western blotting, histological analyses (H&E staining), ELISA assays, and flow cytometry were performed using three randomly selected animals per group (n = 3 biological replicates). For ELISA assays specifically, each biological sample was measured in duplicate (technical replicates). For in vitro experiments, Western blotting and qPCR analyses were performed with three independent cell culture experiments (n = 3 biological replicates), while CCK-8 assays were performed with technical replicates within each experiment. Data are reported as mean ± standard deviation (SD).

All analyses were conducted in the R environment. $P < 0.05$ was considered statistically significant.

## 3. Results

### 3.1 Identification of differentially expressed genes in HF

To explore transcriptional alterations in HF, differential expression analysis was performed using the GSE57345 dataset. Compared with control samples, a large number of genes were dysregulated in HF hearts. For instance, MNS1, FREM1,

and PDE5A were significantly upregulated, whereas SERPINA3, TUBA3D, and FCN3 were downregulated in HF tis-sues. The volcano plot highlighted the global distribution of DEGs (Fig 1A). Heatmap clustering further confirmed distinct expression patterns between control and HF groups, demonstrating that these DEGs could robustly stratify the two cohorts (Fig 1B). These findings indicate that widespread transcriptional reprogramming occurs in HF myocardium.

### 3.2 Functional enrichment analysis of DEGs

To dissect the biological implications of the DEGs, KEGG pathway enrichment analysis was conducted. The results revealed significant enrichment in both metabolic and immune-related pathways, including PI3K–Akt signaling, cytokine–cytokine receptor interaction, complement and coagulation cascades, and multiple metabolic pathways (Fig 2A). Further visualization highlighted a network of metabolic processes such as ketone body metabolism, amino acid metabolism, and folate metabolism (Fig 2B). In parallel, single-sample GSEA (ssGSEA) confirmed widespread activation of metabolic pathways, including lysine degradation, beta-alanine metabolism, histidine metabolism, bile acid biosynthesis, and lipoic acid metabolism, all of which were upregulated in HF (Fig 2C). Gene Ontology (GO) analysis further implicated processes such as cell adhesion, mononuclear cell differentiation, and response to external stimuli (Fig 2D). Together, these results underscore the dual contribution of metabolic dysregulation and immune imbalance to HF pathology.

### 3.3 Upregulation of metabolism-associated pathways in HF

Given the prominence of metabolic signaling in enrichment results, we further examined the expression patterns of genes in key HF-related metabolic pathways. Heatmaps revealed that bile acid biosynthesis (e.g., CYP39A1, CYP46A1),

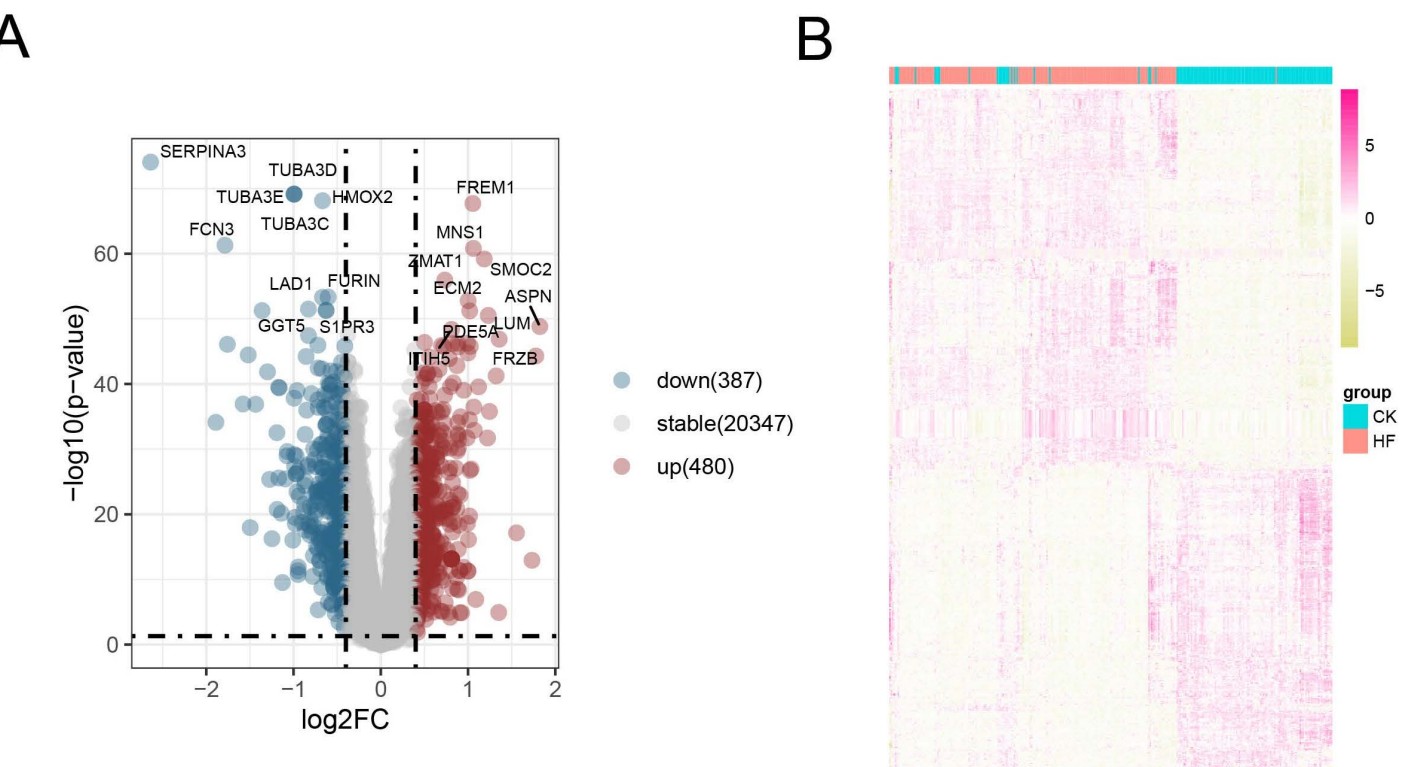

**Fig 1. Identification of differentially expressed genes (DEGs) in the heart failure (HF) dataset GSE57345.** (A) Volcano plot of differential transcriptome analysis ($P < 0.05$, |log2FC| > 0.4). (B) Heatmap showing hierarchical clustering and expression patterns of DEGs in control and HF groups.

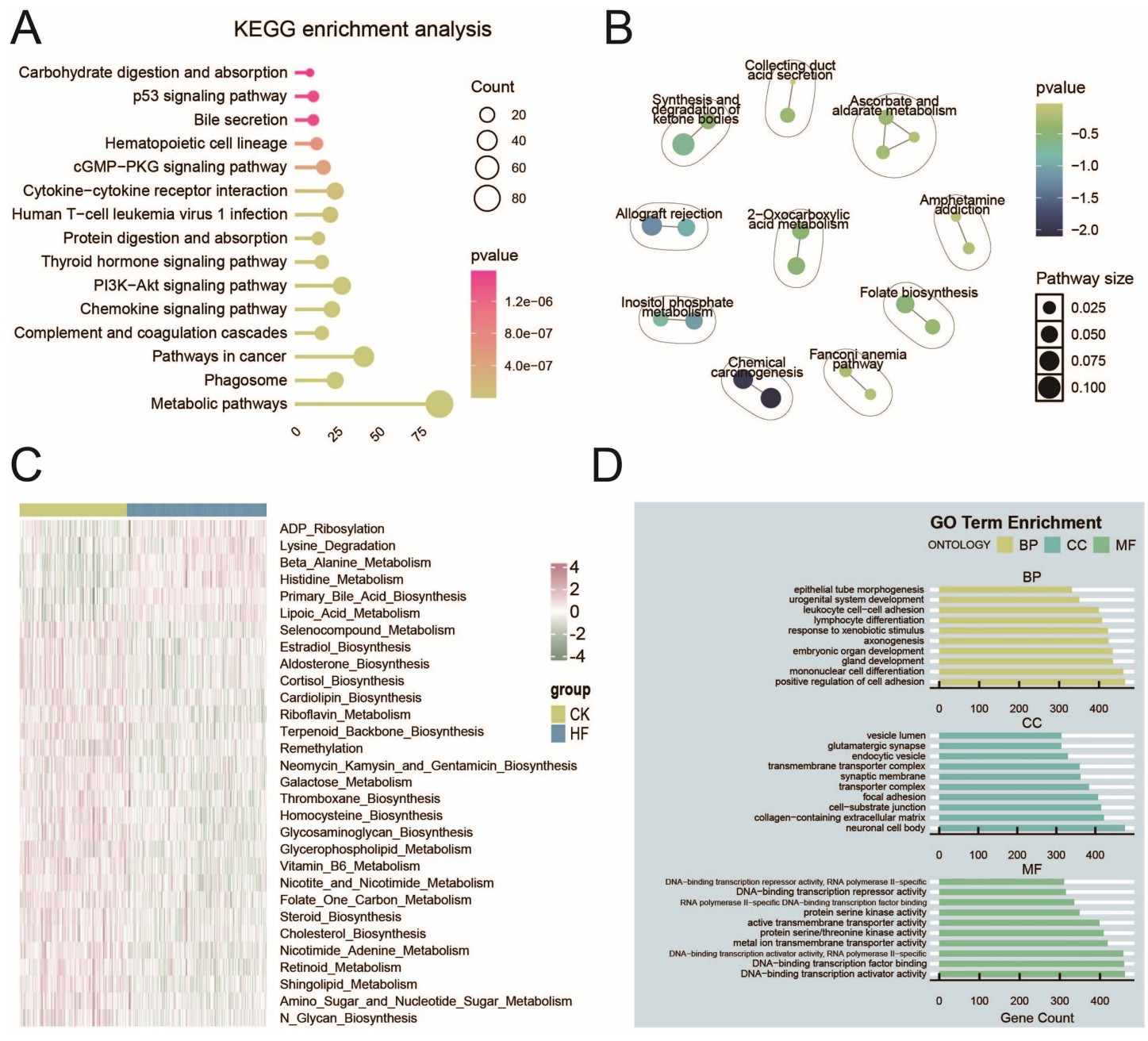

**Fig 2. Enrichment analysis of DEGs in HF.** (A) KEGG pathway enrichment analysis. (B) Functional enrichment network visualization. (C) ssGSEA results of metabolic KEGG pathways. (D) GO enrichment analysis showing representative biological processes.

beta-alanine metabolism, lipoic acid metabolism, histidine metabolism, and lysine degradation pathways were broadly upregulated in HF hearts compared with controls (Fig 3A–E). These changes suggest enhanced metabolic reprogramming in failing myocardium, potentially reflecting altered energy demands and systemic metabolic adaptation in the HF state.

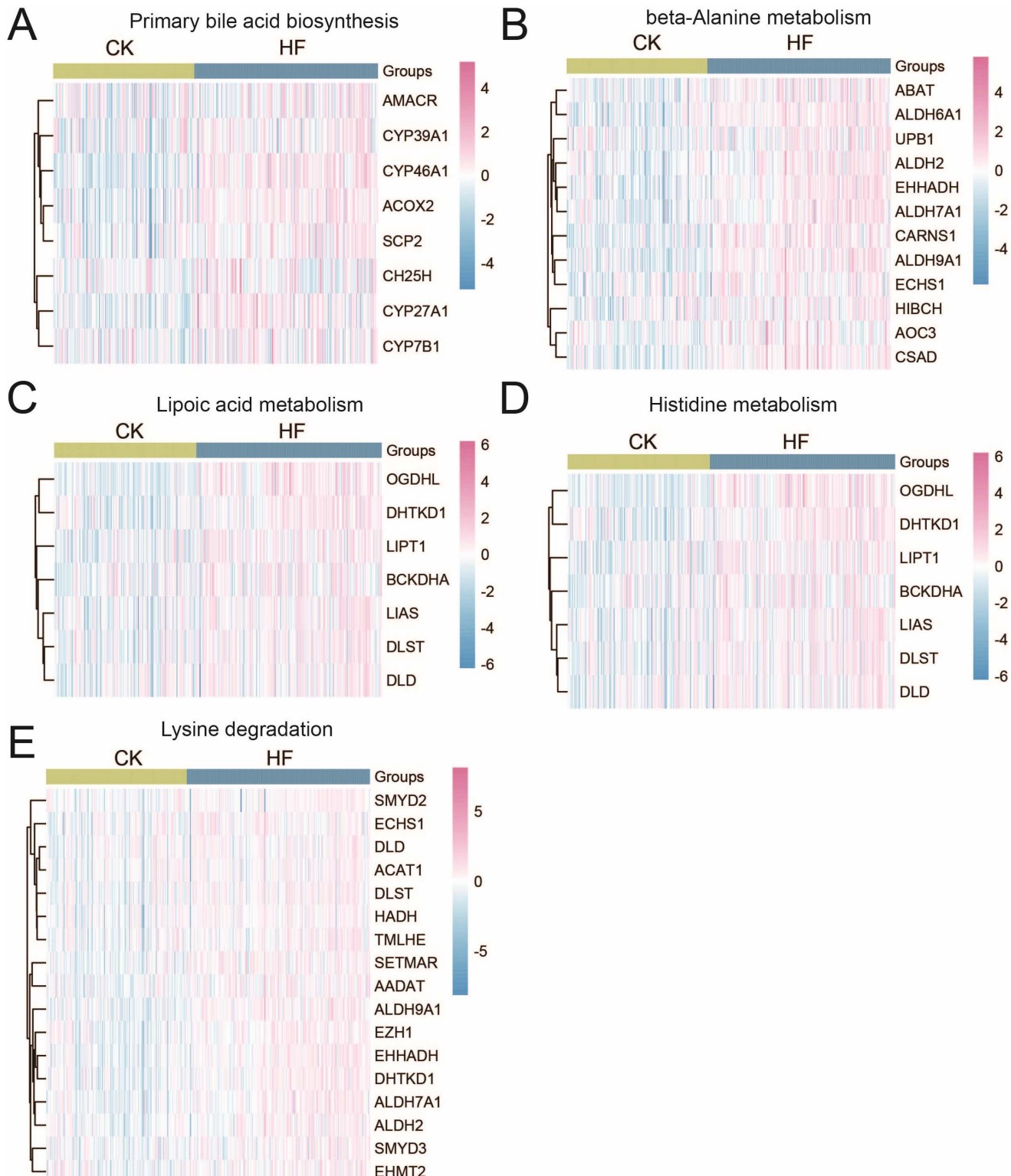

**Fig 3. Expression profiles of HF-related metabolic pathways.** Heatmaps showing gene expression levels involved in (A) primary bile acid biosynthesis, (B) beta-alanine metabolism, (C) lipoic acid metabolism, (D) histidine metabolism, and (E) lysine degradation pathways.

### 3.4 WGCNA identifies STAT1-related immune modules in HF

To investigate co-expression patterns and identify hub genes, weighted gene co-expression network analysis (WGCNA) was applied to the GSE57345 dataset. Several modules were identified, with the ME-CYAN module showing the strongest positive correlation with HF (r = 0.7) (Fig 4A–C). Functional enrichment of this module indicated significant involvement in allograft rejection, PI3K–Akt signaling, chemokine signaling, and ECM–receptor interaction pathways (Fig 4D). Protein–protein interaction (PPI) network analysis further highlighted STAT1, CXCL10, CCL5, MX1, and DDX58 as central hub genes within this module (Fig 4E). KEGG enrichment of these hub genes confirmed strong associations with immune- and inflammation-related pathways such as RIG-I–like receptor, Toll-like receptor, and TNF signaling (Fig 4F). Collectively, these findings indicate that STAT1 and its co-expressed partners may serve as immune regulators central to HF progression.

### 3.5 STAT1 is a potential biomarker and classifier for HF

Expression analysis revealed that the ten hub genes identified by WGCNA were significantly upregulated in HF compared with controls (P < 0.01) (Fig 5A). Receiver operating characteristic (ROC) analysis showed strong discriminative power of several genes, with IFIT3 and MX1 yielding AUC values of 0.869 and 0.829, respectively (Fig 5B). Importantly, random forest analysis ranked STAT1 as the top contributor for distinguishing HF from control samples (Fig 5C). Combined with its central position in the WGCNA co-expression network and its known role in coordinating inflammatory and metabolic pathways, these findings suggest that STAT1 serves not only as a robust potential biomarker for HF classification but also as a key regulatory molecule warranting further mechanistic investigation in HF pathogenesis.

### 3.6 Distinct immune microenvironment remodeling in HF

To further evaluate immune dysregulation, CIBERSORT-based immune infiltration analysis was performed. Marked compositional differences were observed between HF and control groups (Fig 6A–B). Specifically, HF samples exhibited significantly elevated proportions of resting mast cells, plasma cells, macrophages M0 and M1, and CD8+T cells, while naive B cells, macrophages M2, and regulatory T cells were reduced (Fig 6C). These results suggest that HF is characterized by pro-inflammatory immune remodeling, featuring enhanced M1 polarization and CD8+T cell activation, together with impaired regulatory and reparative responses. Such immune imbalance is consistent with our identification of STAT1 as a pro-inflammatory regulator in HF.

### 3.7 Single-cell analysis reveals imbalance of macrophage subpopulations in failing hearts

To further validate the immune infiltration analysis and characterize the immune microenvironment of failing hearts under metabolic stress, we performed computational re-analysis of publicly available single-cell transcriptomic data from human myocardial tissues (GSE145154 dataset) (Control, n = 4; DCM, n = 8; ICM_MI, n = 6; ICM_NMI, n = 6) (Fig 7A–7G). UMAP clustering identified 10 major cell populations, including EC, EndoC, CM, SMC, Pericyte, FB, Myeloid, T, B, and NK cells (Fig 7A). Dotplot analysis confirmed the expression profiles of canonical marker genes for each population (Fig 7B). Comparison of the overall cellular composition among groups revealed significant shifts in immune cell fractions associated with HF progression (Fig 7C).

Focusing on the myeloid compartment (Fig 7D), secondary clustering identified four macrophage subpopulations: M0, M1, M2, and Cycling myeloid (Fig 7E). Marker gene expression patterns (Fig 7F) revealed that the M1 cluster was enriched for inflammation-associated genes (e.g., LST1, NLRP3, KLF4), whereas the M2 cluster was enriched for repair- and metabolism-related genes (e.g., AIF1, S100A9, MS4A4A, FABP5). The M0 cluster displayed homeostatic features, whereas the Cycling myeloid subgroup was characterized by cell-cycle–associated gene expression.

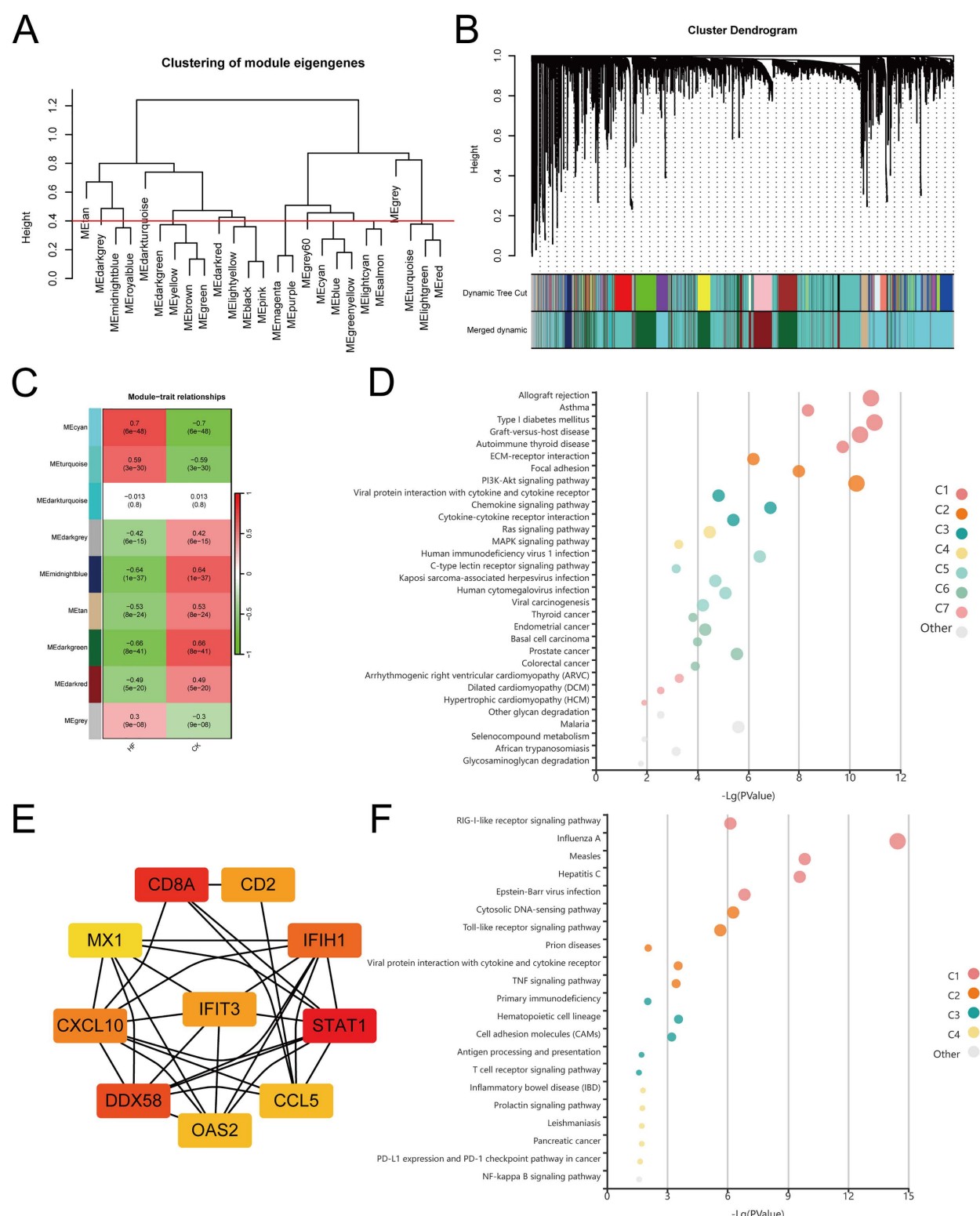

**Fig 4. Hub genes identified using WGCNA.** (A) Clustering dendrogram of module eigengenes.(B) Merged dynamic tree cut showing distinct modules. (C) Module–trait correlation heatmap. (D) KEGG enrichment of the ME-CYAN module. (E) PPI network of ME-CYAN hub genes. (F) KEGG enrichment analysis of hub genes.

**Fig 5. Identification of candidate biomarkers for HF.** (A) Boxplots of hub gene expression in control and HF samples. * $P<0.05$, ** $P<0.01$, *** $P<0.001$. (B) ROC curves assessing the diagnostic performance of hub genes. (C) Random forest ranking showing STAT1 as the top classifier.

Comparison of macrophage subsets across clinical groups (Fig 7G) indicated that M0 cell proportions were relatively stable, while the M1 population showed an increasing trend in DCM and ICM (both infarcted and non-infarcted regions). Conversely, M2 fractions decreased accordingly. These findings, highly consistent with the CIBERSORT-based inference, suggest that under metabolic stress, the myocardial immune microenvironment is skewed toward a pro-inflammatory state

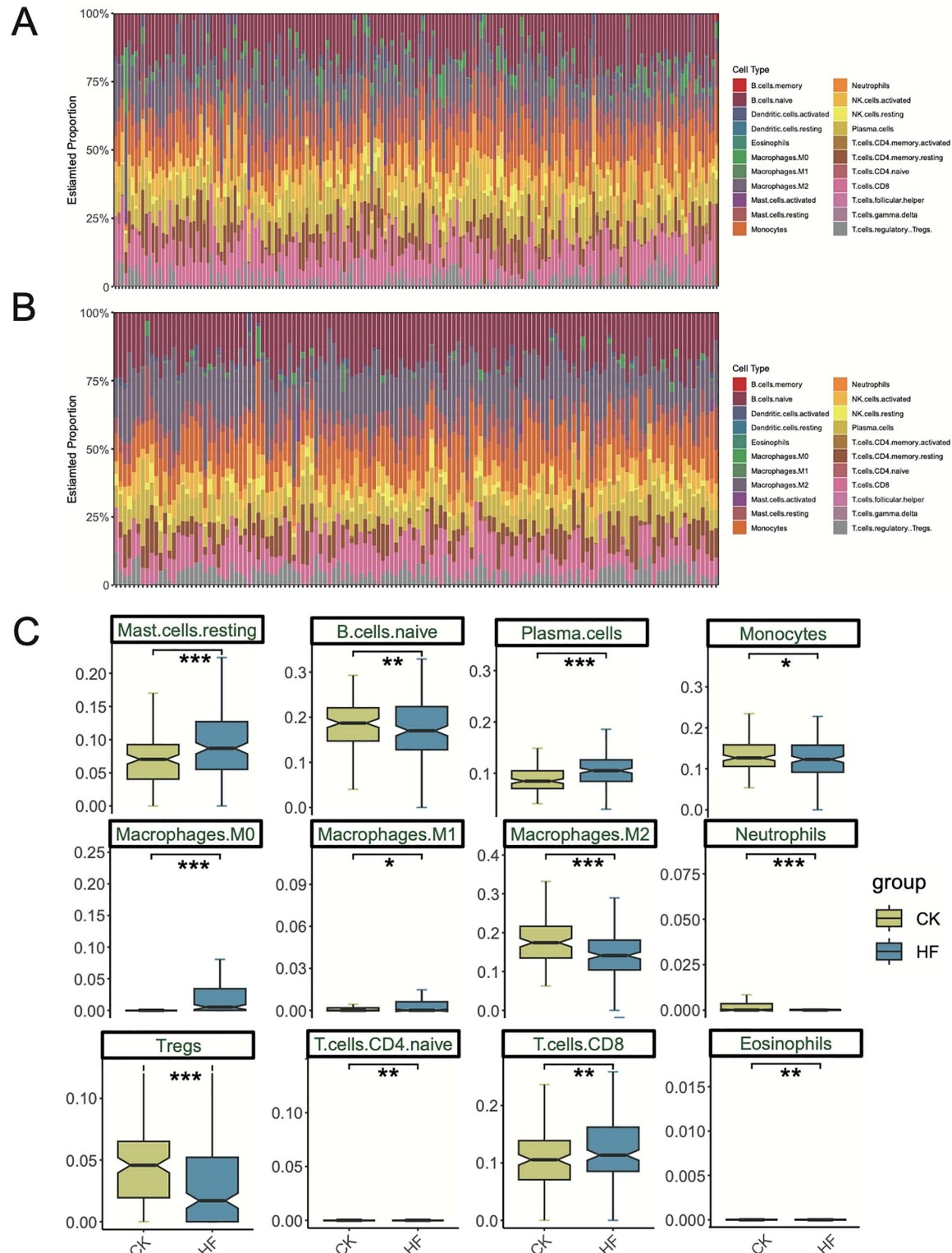

**Fig 6. Immune cell infiltration analysis in HF and control groups.** (A) Distribution of immune cell types in the HF group. (B) Distribution of immune cell types in the control group. (C) Boxplots showing immune cell subsets with significant differences between groups. *P < 0.05, **P < 0.01, ***P < 0.001.

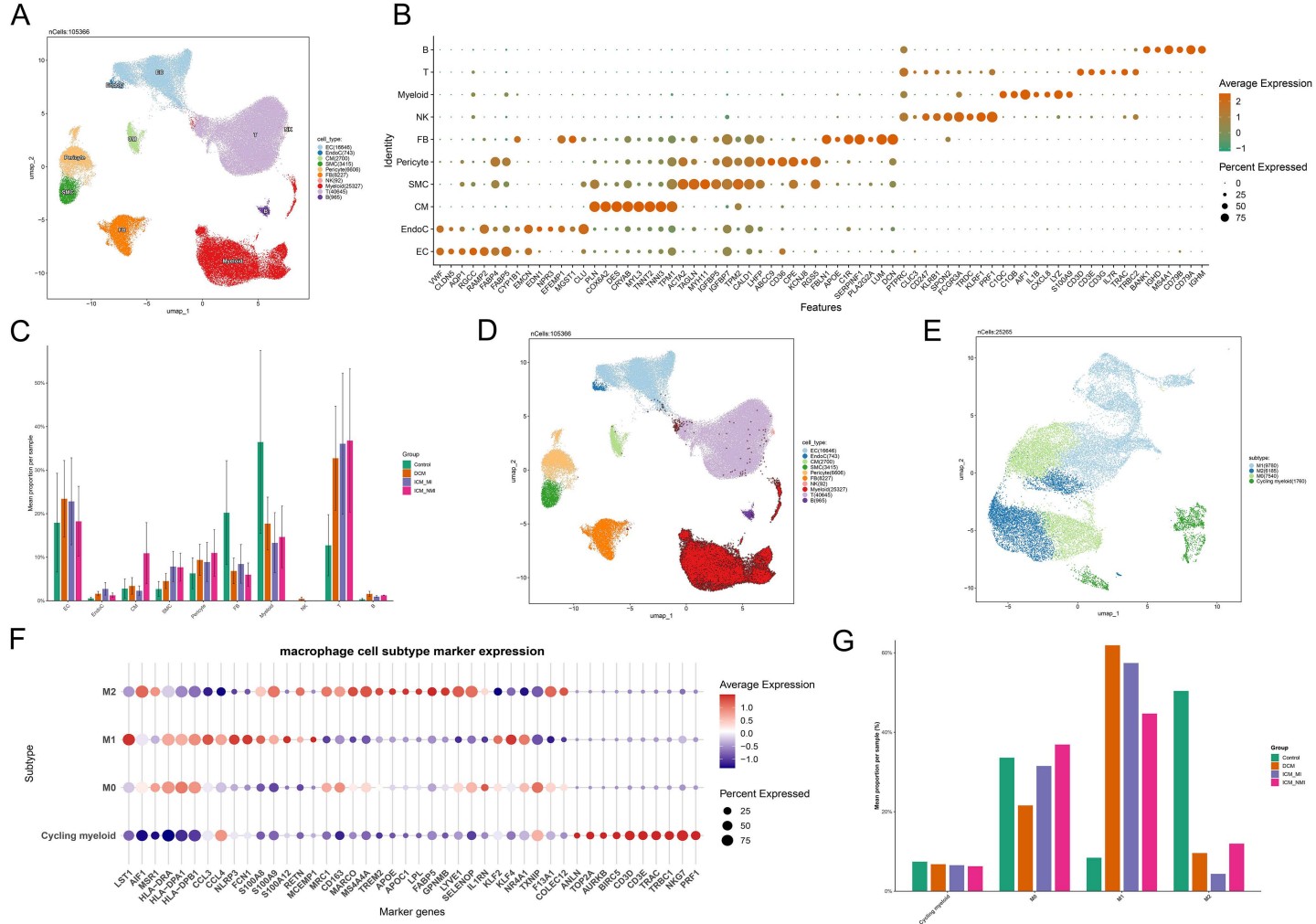

**Fig 7. Single-cell transcriptomic analysis reveals altered macrophage subset composition in failing hearts.** (A) UMAP of major cell populations. (B) Dotplot of canonical marker gene expression. (C) Proportion of cell types across clinical groups (Control, DCM, ICM_MI, ICM_NMI). (D–E) UMAP and clustering of myeloid cells identifying M0, M1, M2, and Cycling myeloid subsets. (F) Marker gene expression patterns of each macrophage subset. (G) Distribution of M0/M1/M2 subsets across groups.

characterized by elevated M1 and reduced M2 macrophages. Together with transcriptomic evidence of STAT1 pathway activation, our single-cell analysis highlights the potential role of STAT1 in regulating macrophage polarization in the pathological process of HF under metabolic stress.

### 3.8 Progressive suppression of JAK–STAT signaling in failing myocardium, most prominent in ischemic HF

To further explore signaling pathway alterations under the context of metabolic stress, we calculated enrichment scores using the publicly available single-cell RNA-seq data from the GSE145154 dataset, with a specific focus on interferon-associated pathways and JAK–STAT signaling across different cell populations (Fig 8A–H). In all major cell types, IFNA and IFNG pathway activity scores were highest in the Control group, decreased in DCM, and further declined in ICM (both infarcted and non-infarcted regions) (Fig 8A–D). Similarly, JAK and JAK–STAT scores exhibited a stepwise reduction (Fig 8E–F), indicating a close association between HF and suppression of immune signaling.

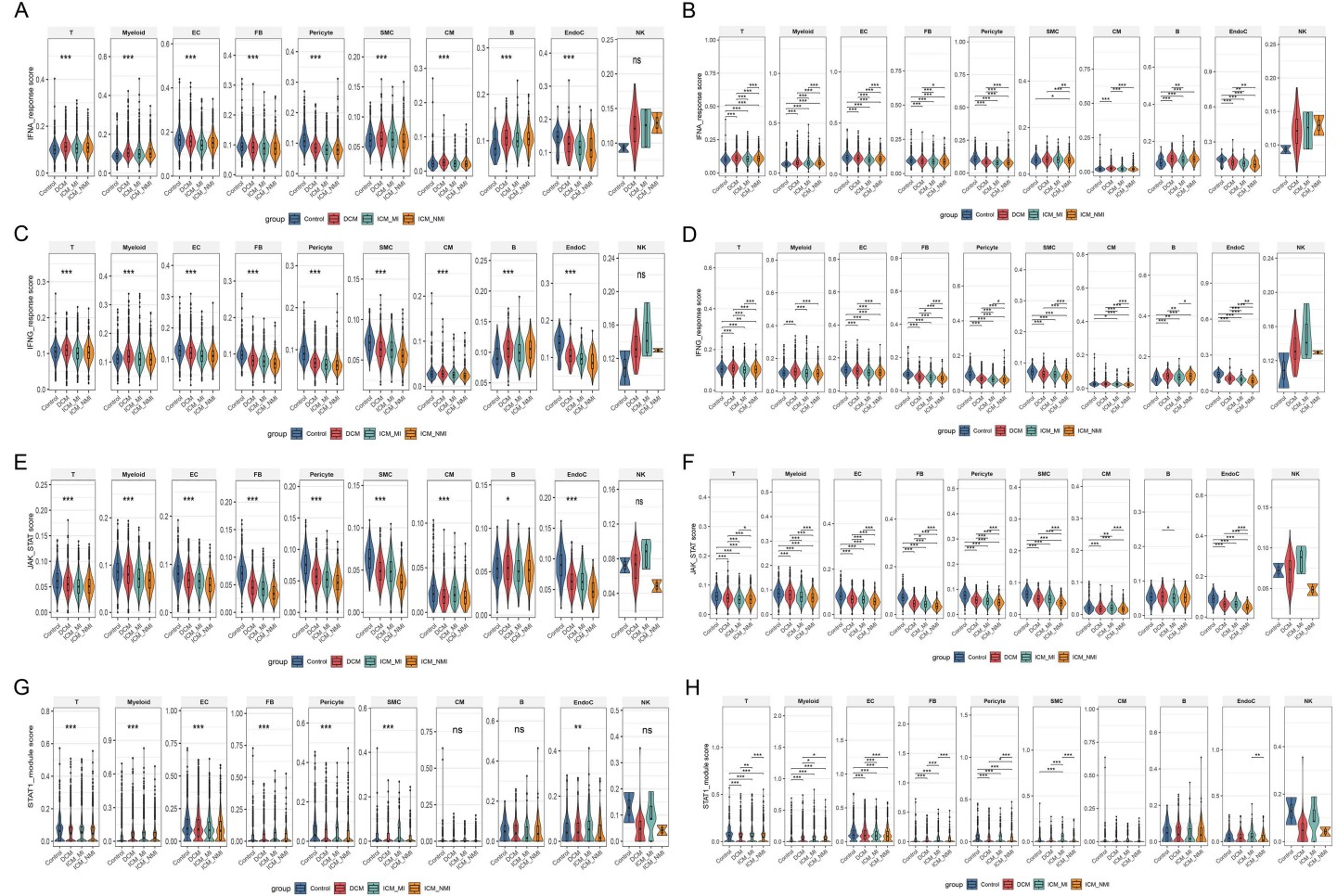

**Fig 8. Single-cell transcriptomics highlight alterations in signaling pathways across cell types in failing myocardium.** (A–B) IFNA pathway scores and group comparisons. (C–D) IFNG pathway scores and group comparisons. (E–F) JAK and JAK–STAT pathway scores and group comparisons. (G–H) STAT1-related scores and group comparisons. *P < 0.05, **P < 0.01, ***P < 0.001.

Further analysis of STAT1 activity scores demonstrated consistent declines across multiple cell types (Fig 8G–H), with the most pronounced reduction observed in ICM tissues. These findings align with our CIBERSORT and single-cell subpopulation analyses, underscoring STAT1 signaling as a crucial factor in immune microenvironment remodeling during HF. Based on this observation, we selected ICM samples for subsequent experiments, given the most marked signaling alterations in this group.

### 3.9 Decline of JAK–STAT pathway but elevated STAT1 module in macrophage subsets

At the macrophage subset level, IFN- and JAK–STAT-associated activity was further evaluated (Fig 9A–H). The results showed no significant overall differences in IFNA or IFNG pathway scores among the M0, M1, M2, and Cycling myeloid subsets (Fig 9A–D), suggesting that interferon signaling was not systematically downregulated at the subset level. In contrast, JAK and JAK–STAT pathway scores decreased across all subsets (Fig 9E–F), with more pronounced reductions in ICM samples, indicating pathway suppression in the failing heart.

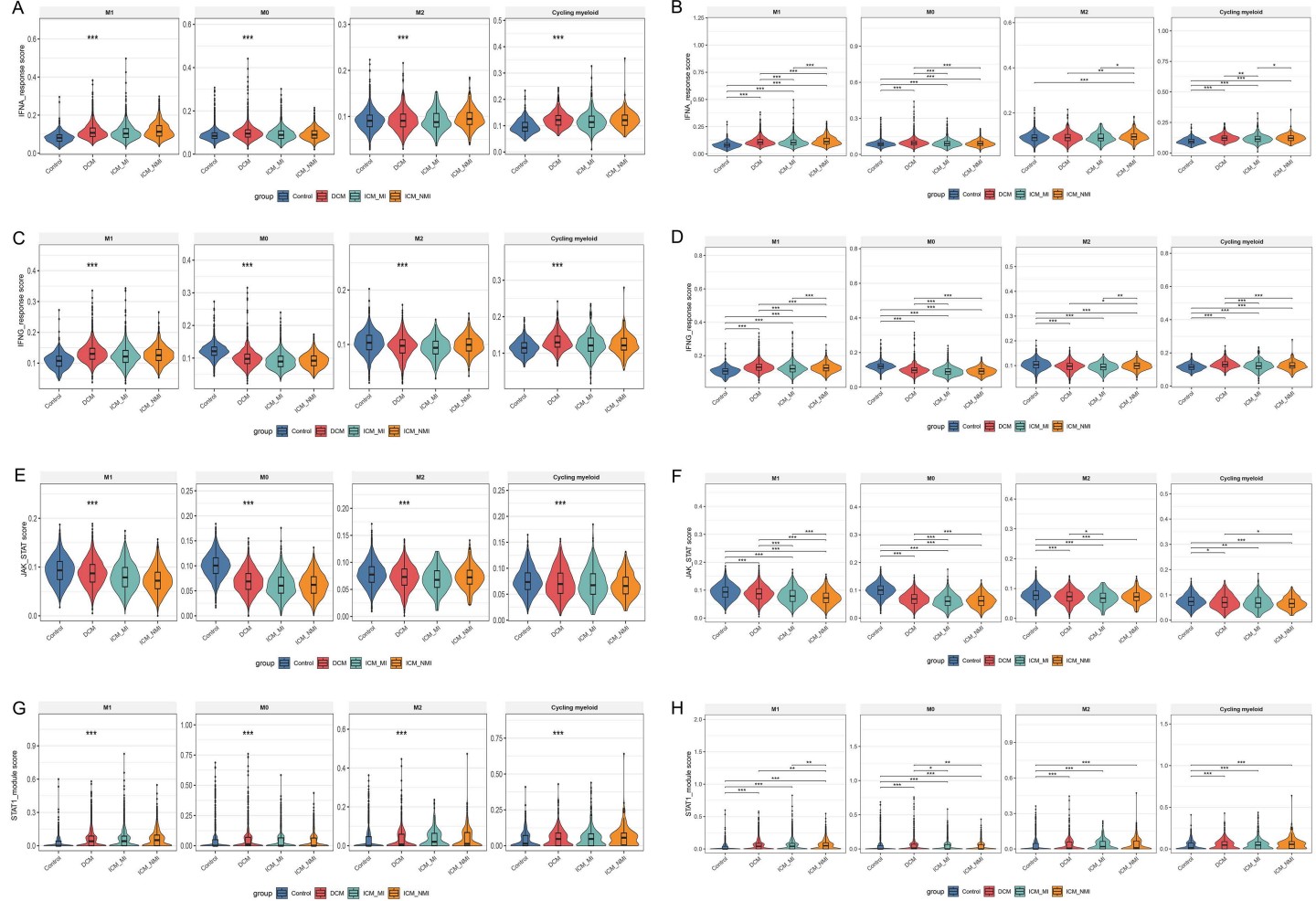

**Fig 9. Single-cell transcriptomic scoring of signaling pathways in macrophage subsets.** (A–B) IFNA pathway scores in M0, M1, M2, and Cycling myeloid subsets. (C–D) IFNG pathway scores and group comparisons. (E–F) JAK and JAK–STAT pathway scores and group comparisons. (G–H) STAT1 module scores and group comparisons. *P<0.05, **P<0.01, ***P<0.001.

Notably, STAT1 module scores increased in all macrophage subsets compared with the Control group (Fig 9G–H), particularly in M1 and Cycling myeloid clusters. This paradoxical pattern suggests that although overall JAK–STAT signaling was attenuated, downstream STAT1 may undergo aberrant activation in specific macrophage subsets, fueling inflammatory responses.

### 3.10 DAPA alleviated myocardial injury and inflammatory response in rats with HF

To assess the cardioprotective effects of dapagliflozin, histological and molecular evaluations of myocardial tissue were performed. In the control (CK) group, myocardial fibers were regularly arranged with intact nuclei, uniform staining, and no signs of necrosis or inflammatory cell infiltration. In contrast, hearts from the HF group displayed highly disorganized fibers accompanied by rupture, dissolution, and marked inflammatory infiltration. DAPA administration markedly improved tissue architecture, showing better fiber alignment, strengthened inter-fiber connections, and reduced inflammatory infiltration, indicating attenuation of myocardial injury (Fig 10A). Western blot analysis further demonstrated that STAT1 protein

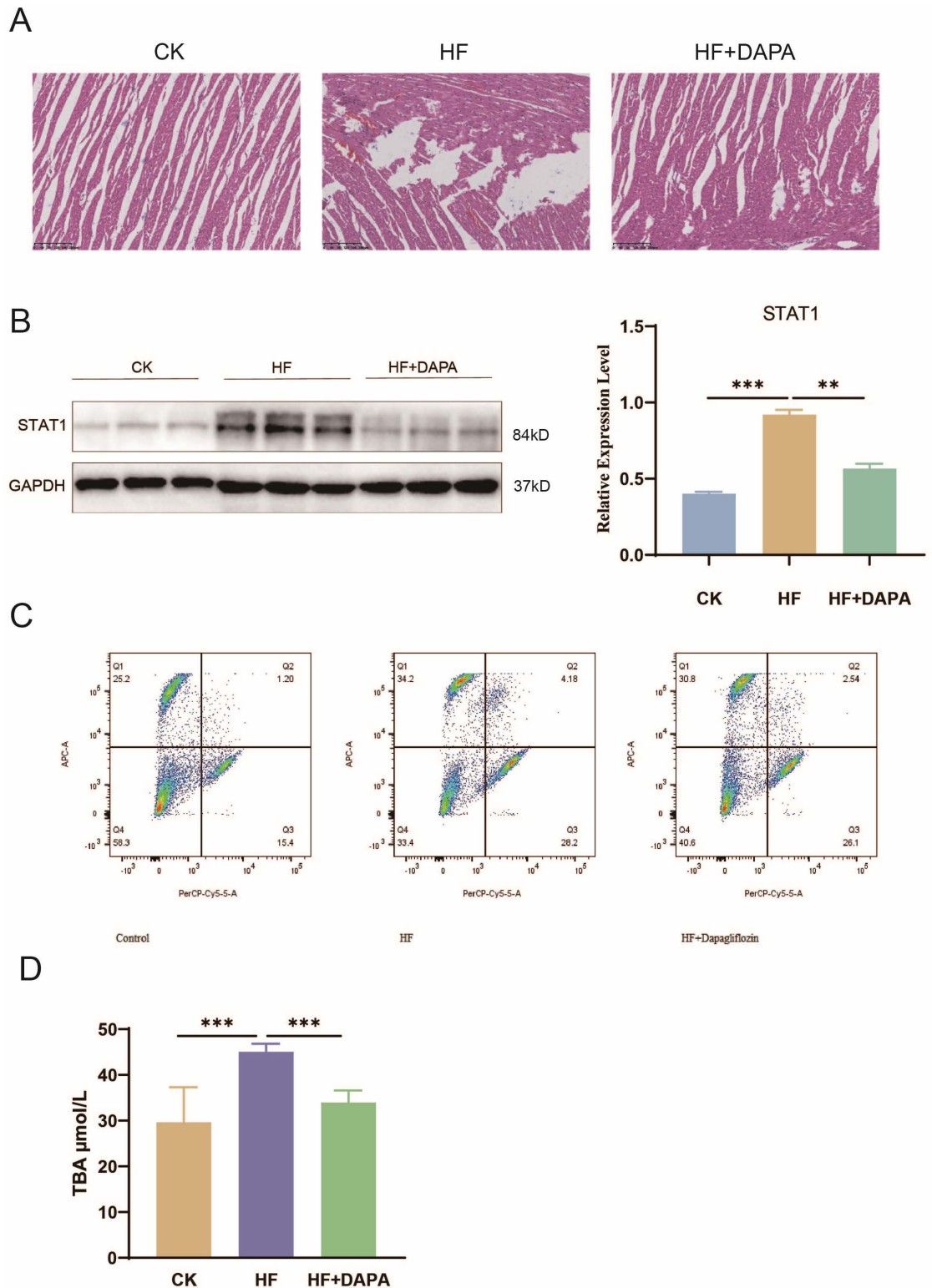

**Fig 10. Histological and molecular evidence of the cardioprotective effects of dapagliflozin in HF rats.** (A) Representative H&E staining of myocardial sections from CK, HF, and HF + DAPA groups showing tissue architecture and inflammatory infiltration changes. (B) Myocardial STAT1 protein

levels detected by Western blot. (C) Flow cytometric analysis of macrophage polarization (M1: CD86+, M2: CD163+). (D) Serum total bile acid (TBA) concentrations in different groups. Data are presented as mean±SD. Statistical analysis was performed using unpaired two-tailed Student's t-test; *P<0.05, **P<0.01, ***P<0.001.

expression was significantly increased in HF myocardium, whereas DAPA substantially reduced STAT1 protein levels (Fig 10B), supporting the involvement of STAT1 downregulation in its protective action.

Flow cytometric analysis of peripheral blood revealed that DAPA significantly lowered the proportion of pro-inflammatory M1 macrophages while partially restoring M2 macrophage levels, consistent with modulation of immune homeostasis (Fig 10C). In line with this, serum concentrations of IL-6 and TNF-α were markedly elevated in HF rats but sharply declined after DAPA treatment, confirming its anti-inflammatory activity (S1B Fig). Moreover, consistent with our bioinformatics findings showing activation of bile acid biosynthesis pathways in HF (Fig 3A), HF rats exhibited increased total bile acid (TBA) levels, which were effectively reduced by DAPA (Fig 10D), suggesting amelioration of bile acid metabolism dysregulation. Collectively, these findings demonstrate that DAPA counteracts HF-induced pathological changes via complementary mechanisms, including inhibition of STAT1 signaling, regulation of macrophage polarization, suppression of pro-inflammatory cytokine production, reduction of bile acid accumulation, and improvement of myocardial tissue architecture.

### 3.11 DAPA blocked STAT1 expression to attenuate myocardial cell injury

Considering bioinformatics evidence and in vivo results, STAT1 emerged as a potential biomarker of HF and a key target of DAPA. To confirm this, STAT1-overexpressing H9C2 cardiomyocytes were generated (Fig 11A). CCK-8 assays showed that STAT1 overexpression significantly reduced cell viability (Fig 11B), while flow cytometry revealed increased early, late, and overall apoptotic fractions (Fig 11C). Strikingly, treatment with DAPA greatly suppressed STAT1 expression in STAT1-overexpressing cells (Fig 11D), rescued cell viability (Fig 11E), and reduced apoptotic rates (Fig 11F).

These in vitro findings reinforce the in vivo results, demonstrating that DAPA protects cardiomyocytes by suppressing STAT1 protein expression, thereby alleviating apoptosis and restoring cellular viability. Taken together, the data confirm that STAT1 downregulation constitutes an important molecular mechanism underlying the cardioprotective effects of DAPA (S2 Fig).

## 4. Discussion

In this study, we integrated bioinformatics analyses, single-cell transcriptomics, and experimental validation to elucidate the role of STAT1 in HF and to investigate the therapeutic mechanisms of DAPA. Our data revealed that STAT1 is a central regulator linking metabolic stress and pro-inflammatory immune remodeling in HF. Importantly, we demonstrated that DAPA treatment is associated with reduced STAT1 protein expression in HF, which correlates with modulation of macrophage polarization, reduction of inflammatory cytokine production, and mitigation of cardiac remodeling.

### STAT1 as a driver of metabolic and immune dysregulation in HF

Our bulk transcriptome and WGCNA analyses identified STAT1 as one of the hub factors most strongly associated with HF. Differential expression and classification analyses confirmed that STAT1 expression was significantly elevated in HF patients, highlighting its potential as a biomarker. Mechanistically, HF was characterized by upregulation of metabolic stress pathways—including bile acid biosynthesis, amino acid metabolism, and lipoic acid metabolism—paralleled by reconfiguration of the immune microenvironment with decreased M2-type and increased M1-type macrophages. These findings are consistent with reports that metabolic disturbances and chronic inflammation act synergistically to drive myocardial injury and remodeling [8,9,36]. As STAT1 is established as a transcriptional regulator of interferon signaling and

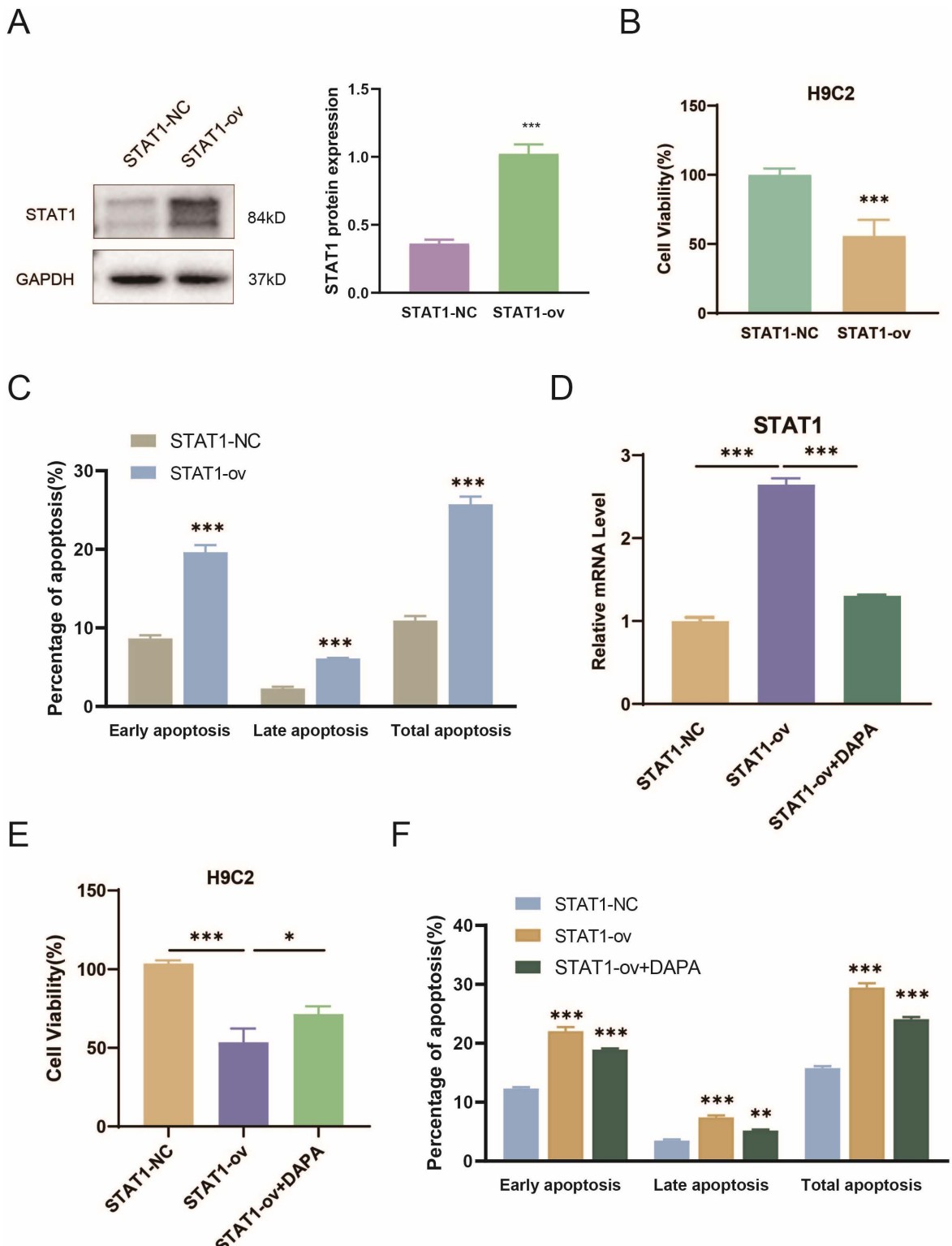

**Fig 11. Dapagliflozin reduces STAT1 expression and alleviates myocardial cell injury in vitro.** (A) Verification of STAT1 overexpression in H9C2 cells. (B) Effects of STAT1 overexpression on H9C2 cell viability (CCK-8 assay). (C) Flow cytometric analysis of apoptosis in STAT1-overexpressing

cells. (D) STAT1 protein expression after DAPA treatment in STAT1-overexpressing cells. (E) Rescue of cell viability by DAPA. (F) Reduction of apoptotic cell fractions following DAPA treatment. Data are shown as mean±SD. Statistical comparisons were conducted using unpaired two-tailed Student's t-test; *P<0.05, **P<0.01, ***P<0.001.

inflammatory apoptosis [10,11,37], our data extend its pathogenic role by linking elevated STAT1 expression with metabolic stress–induced immune imbalance in HF.

### Single-cell analysis highlights STAT1-driven macrophage polarization

Using single-cell transcriptomic analysis, we further demonstrated that HF tissues harbor distinct shifts in macrophage subpopulations. M1 macrophages—enriched for pro-inflammatory genes—were expanded, while reparative M2 macrophages were reduced across HF groups. Consistently, STAT1 signaling scores were elevated in macrophage subsets, especially in M1 and proliferating myeloid clusters, despite an overall suppression of upstream JAK–STAT signaling. This apparent paradox suggests that STAT1 can bypass global JAK–STAT suppression, becoming aberrantly activated in selected immune compartments to reinforce inflammatory polarization. These findings emphasize the relevance of single-cell profiling in dissecting heterogeneity that cannot be captured by bulk analysis, and provide insights into how STAT1 shapes the immune landscape of failing myocardium.

### Dapagliflozin treatment is associated with reduced STAT1 expression in HF

DAPA treatment significantly ameliorated myocardial injury and inflammatory infiltration in our rat HF model. In vivo, DAPA suppressed myocardial STAT1 expression, reduced circulating pro-inflammatory cytokines (IL-6 and TNF-α), and restored the balance between M1 and M2 macrophages. In vitro, DAPA rescued STAT1-overexpressing cardiomyocytes from enhanced apoptosis and impaired viability. These findings suggest STAT1 expression may be associated with DAPA's cardioprotective effects, though whether STAT1 is a direct pharmacological target or an indirect consequence of DAPA treatment requires further investigation. Previous studies reported the pleiotropic benefits of SGLT2 inhibitors, including improvement of cardiac energetics, reduction of oxidative stress, and suppression of NLRP3 inflammasome activation [21–23]. Our study complements these observations by identifying STAT1 suppression as a novel mechanism underlying the cardioprotective actions of DAPA, independent of its glucose-lowering effects.

The robust evidence of a class effect for SGLT2 inhibitors in reducing HF hospitalization and cardiovascular mortality [38,39] suggests that the fundamental mechanisms, such as STAT1 inhibition and subsequent immune/metabolic modulation, may be shared across the class. Future studies directly comparing the effects of different SGLT2 inhibitors (e.g., Dapagliflozin vs. Empagliflozin) on the STAT1 pathway are warranted to definitively confirm this proposed class effect.

### Clinical and translational implications

Taken together, our findings suggest STAT1 may serve as a biomarker associated with HF progression and DAPA response in HF. Elevated STAT1 expression may serve as an indicator of heightened metabolic stress and pro-inflammatory activity, offering potential diagnostic and prognostic value. Furthermore, the association between DAPA treatment and reduced STAT1 protein expression suggests that patient stratification based on STAT1 expression levels could help optimize therapeutic response. These insights may also inform the design of combinatorial treatment strategies, for example, the addition of STAT1-targeted agents to established HF regimens [40–43].

### Limitations and future directions

Several limitations should be acknowledged. First, although single-cell analysis provided valuable insights into immune and metabolic heterogeneity, our dataset was retrospective and limited in scale. Larger multi-omics studies, particularly

those integrating metabolomics and proteomics, are required to validate these findings. Second, our Western blot data measured total STAT1 protein expression rather than phosphorylation status (p-STAT1 at Tyr701/Ser727) or transcriptional activity. Future studies should include phospho-STAT1 Western blotting, nuclear/cytoplasmic fractionation, STAT1 target gene expression analysis, and ChIP assays to directly assess STAT1 expression. Third, our findings do not establish STAT1 as a direct pharmacological target of DAPA; STAT1 reduction could be an indirect consequence of DAPA's effects on metabolic normalization or oxidative stress reduction. STAT1 loss-of-function studies and drug-target engagement assays would be valuable to establish causality. Additionally, this study exclusively used male rats to minimize hormonal variability associated with the estrous cycle, which could confound infarct size and cardiac remodeling during model establishment. However, given well-documented sex differences in heart failure pathophysiology—including patterns of hypertrophy, fibrosis, and drug metabolism—our findings should not be directly extrapolated to female subjects and require validation in sex-balanced experimental designs. Finally, the translational applicability of our findings should be confirmed in prospective clinical cohorts of HF patients receiving DAPA.

## 5. Conclusions

In summary, this study reveals STAT1 as a critical mediator of metabolic stress and immune dysregulation in heart failure, and identifies its suppression as a novel cardioprotective mechanism of dapagliflozin. By bridging transcriptome analyses with functional experiments, we provide mechanistic evidence that DAPA mitigates HF progression through suppression of STAT1 expression and immune microenvironment modulation. These findings not only enhance our understanding of HF biology but also suggest STAT1 as a potential biomarker associated with DAPA's cardioprotective effects for precision management of HF.

## Supporting information

**S1 Fig. Evaluating myocardial histopathology.** (A) and serum inflammatory cytokine levels (IL-6 and TNF-α) (B) in control (CK), heart failure (HF), and HF rats treated with dapagliflozin at 3, 6, or 9 mg/kg/day.
(TIFF)

**S2 Fig. Raw images for the experimental section.**
(PDF)

## Author contributions

Conceptualization: Lianqi He.

Data curation: Lianqi He.

Formal analysis: Di Zhang.

Funding acquisition: Di Zhang.

Investigation: Di Zhang, Zhanchun Song.

Methodology: Shiwen Zhang.

Project administration: Shiwen Zhang, Zhichao Ren.

Resources: Shiwen Zhang, Zhichao Ren, Xiran Cheng, Zhanchun Song.

Software: Zhichao Ren.

Supervision: Zhichao Ren, Xiran Cheng.

Validation: Yihong Huangfu.

Visualization: Yihong Huangfu, Xiran Cheng.

Writing – original draft: Yihong Huangfu.

Writing – review & editing: Zhanchun Song.

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
