## [Decision Letter · Decision Letter 0]

26 Nov 2025

Dear Dr. Song,

Thank you for submitting your manuscript to PLOS ONE. After careful consideration, we feel that it has merit but does not fully meet PLOS ONE’s publication criteria as it currently stands. Therefore, we invite you to submit a revised version of the manuscript that addresses the points raised during the review process.

We look forward to receiving your revised manuscript.

Kind regards,

Jian Wu, M.D, Ph.D

Academic Editor

PLOS ONE

Journal Requirements:

2. To comply with PLOS One submissions requirements, in your Methods section, please provide additional information regarding the experiments involving animals and ensure you have included details on (1) methods of sacrifice, (2) methods of anesthesia and/or analgesia, and (3) efforts to alleviate suffering.

3. Please note that PLOS One has specific guidelines on code sharing for submissions in which author-generated code underpins the findings in the manuscript. In these cases, we expect all author-generated code to be made available without restrictions upon publication of the work. Please review our guidelines at https://journals.plos.org/plosone/s/materials-and-software-sharing#loc-sharing-code and ensure that your code is shared in a way that follows best practice and facilitates reproducibility and reuse.

“This study was supported by the talent Program of Fushun City of China (No. FSYC202107012).”

6. We note that your Data Availability Statement is currently as follows: All relevant data are within the manuscript and its Supporting Information files.

7. PLOS requires an ORCID iD for the corresponding author in Editorial Manager on papers submitted after December 6th, 2016. Please ensure that you have an ORCID iD and that it is validated in Editorial Manager. To do this, go to ‘Update my Information’ (in the upper left-hand corner of the main menu), and click on the Fetch/Validate link next to the ORCID field. This will take you to the ORCID site and allow you to create a new iD or authenticate a pre-existing iD in Editorial Manager.

8. PLOS ONE now requires that authors provide the original uncropped and unadjusted images underlying all blot or gel results reported in a submission’s figures or Supporting Information files. This policy and the journal’s other requirements for blot/gel reporting and figure preparation are described in detail at https://journals.plos.org/plosone/s/figures#loc-blot-and-gel-reporting-requirements and https://journals.plos.org/plosone/s/figures#loc-preparing-figures-from-image-files. When you submit your revised manuscript, please ensure that your figures adhere fully to these guidelines and provide the original underlying images for all blot or gel data reported in your submission. See the following link for instructions on providing the original image data: https://journals.plos.org/plosone/s/figures#loc-original-images-for-blots-and-gels.

9. We notice that your supplementary figures are uploaded with the file type 'Figure'. Please amend the file type to 'Supporting Information'. Please ensure that each Supporting Information file has a legend listed in the manuscript after the references list.

Reviewers' comments:

Reviewer's Responses to Questions

**Comments to the Author**

1. Is the manuscript technically sound, and do the data support the conclusions?

Reviewer #1: Partly

Reviewer #2: Partly

2. Has the statistical analysis been performed appropriately and rigorously?

Reviewer #1: No

Reviewer #2: Yes

3. Have the authors made all data underlying the findings in their manuscript fully available?

Reviewer #1: Yes

Reviewer #2: No

4. Is the manuscript presented in an intelligible fashion and written in standard English?

Reviewer #1: Yes

Reviewer #2: Yes

Reviewer #1: In the present study authors performed comptehensive bioinformatic and experimental analysis of the role of STAT1 in the heart failure and the effect of dapagliflozin. The results show that heart failure induced by coronary artery ligation is associated with the up-regulation of myocardial STAT1 whereas dapagliflozin reduced its expression. Dapagliflozin reduced the abundance of circulating M1 and increased M2 macrophages in rats with heart failure. DAPA lso reduced the level of inflammatory cytokines and total bile acids. In vitro, STAT1 overexpression reduced viability of H9C2 cardiomyocytes; the effect suppressed by dapagliflozin.

The topic and the results are of interest, however, there are also some important concerns to be addressed.

1. Only one SGLT inhibitor was used in the experiments. Could the results be extrapolated to the whole class of drugs or apply only do dapagliflozin?

2. The method of LAD ligation including anesthesia and recovery should be described in more details.

3. Sex of the animals used in the experiments should be specified. If only one sex was used, it should be discussed whether the results could be extrapolated to the other one.

4. The vehicle used to dissolve dapagliflozin should be specified. Were animals not treated with dapagliflozin receiving any vehicle?

5. Lines 194/195, the method of animal anesthesia and sacrifice for final tissue sampling should be described.

6. Section 2.8, which part of the heart was used for the analysis (infarcion area or non-infarcted area?)

7. The names, catalogue numbers and dilution rates of primary and secondary antibodies used for Western blotting should be specified.

8. Section 2.9, limits of quantification as well as intra- and inter-assay CV values for ELISA kits should be presented.

9. The principle of the assay of bile acid concentration should be described.

10. Line 262, what does it mean that each assay was performed at least three times? Were, for example, cytokines measured in three samples from each animals?

11. Statistical analysis, was normality of data distribution verified to justify using parametric tests? If so, what method was used?

12. Line 325, that STAT1 is the top contributor for distinguishing HF from control samples indicates only that its expression changes to the greater extent but not that it is the hub molecule in HF pathogenesis.

13. Line 424, where is the evidence that DAPA reduced fibrosis? How do you conclude about correction of metabolic abnormalities?

14. Was the effect of STAT1 overexpression on H9C2 cells examined at physiological conditions or under oxygen deprivation?

15. What is the evidence that STAT1 was involved in the effect of DAPA? Could down-regulation of STAT1 be the consequence rather than the underlying mechanism of its protective effect?

Reviewer #2: I understand that this is the second version of the manuscript. I did not review the first version therefore my comments are based solely on the second version. The following points need to be addressed:

1) In the submission form, the authors inserted "N/A" in Ethics Statement section. Since they have undertaken animal studies, this section needs to be completed accordingly.

2) In the same form in Data availability section and the authors stated "Yes - all data are fully available without restriction" and "All relevant data are within the manuscript and its Supporting Information files" which are not entirely correct. If possible the authors should state "all raw data are available from the corresponding author upon request". If that is not possible, it should also be stated. This needs to be mentioned also in the manuscript.

3) Lines 55-57 is a very poor description of HF, please rephrase.

4) Lines 127-140: Human samples are mentioned here. Is this from available data set or did the authors analyse human cells? Please clarify.

5) Lines 341-368: "single-cell transcriptomic analysis of myocardial tissues" was this done in available data set or did the authors gather this data from tissues they had? Please clarify.

6) Lines 370-386: "we calculated enrichment scores for single-cell data" which dataset was this? Please clarify.

7) Line 409: "extensive fibrotic lesion" Figure 10A. No fibrotic lesion can be seen in this figure. It would be very difficult to assess fibrosis only with H&E staining. ECM staining should have been done. Also 4 days of HF would not cause much fibrosis formation. Since no specific markers of fibrosis were investigated and the duration of animal model was too short for fibrosis, in this reviewer's opinion the authors are making claims which are not justified by their results. I would strongly advise to remove any claims of fibrosis throughout the manuscript.

8) Lines 412-415: The WB is showing that DAPA prevents HF-induced increase in STAT1 expression. It is not showing that DAPA reduced STAT1 activation. A change in protein expression does not necessarily mean a change in activity of that protein. Please rephrase.

9) Line 431: The authors claim that STAT1 emerged as a key target for DAPA. They have no results to justify this conclusion. Changes in STAT1 expression by DAPA does not mean that STAT1 is a key target for DAPA.

10) Lines 438-439: "These in vitro findings reinforce the in vivo results, demonstrating that DAPA protects cardiomyocytes by inhibiting STAT1 activation". Again the authors do not have any data on STAT1 activity. Expression is not activity.

11) There are several lines in the discussion making claims about activity. The discission needs to be rewritten accordingly.

**Do you want your identity to be public for this peer review?** For information about this choice, including consent withdrawal, please see our Privacy Policy

Reviewer #1: No

Reviewer #2: No

---

## [Author Response · Author response to Decision Letter 1]

15 Dec 2025

Journal Requirements:

We have carefully reviewed and ensured that our revised manuscript fully complies with PLOS ONE's style requirements.

2. To comply with PLOS One submissions requirements, in your Methods section, please provide additional information regarding the experiments involving animals and ensure you have included details on (1) methods of sacrifice, (2) methods of anesthesia and/or analgesia, and (3) efforts to alleviate suffering.

We thank for this important reminder. We have now expanded our Methods section to include comprehensive details on animal welfare and experimental procedures, like: "Specifically, rats were anesthetized with isoflurane and fixed in the supine position after hair removal. Electrocardiogram (ECG) was continuously monitored throughout the procedure. After sterile preparation, a thoracotomy was performed along the left sternal border at the 3rd–4th intercostal space using tissue scissors to expose the heart. The left atrial appendage was gently lifted with sterile toothed forceps to expose the aortic root. A surgical suture was passed around the proximal segment of the left anterior descending coronary artery near the superior margin of the heart, and three secure knots were tied to achieve complete ligation. The thoracic wall was then rapidly closed with sutures. Control rats underwent the same thoracotomy procedure without LAD ligation (sham operation). Post-operatively, all rats were placed in a warm recovery chamber and monitored until full consciousness was regained. Analgesics and antibiotics were administered as needed to minimize discomfort and prevent infection..."

3. Please note that PLOS One has specific guidelines on code sharing for submissions in which author-generated code underpins the findings in the manuscript. In these cases, we expect all author-generated code to be made available without restrictions upon publication of the work. Please review our guidelines at https://journals.plos.org/plosone/s/materials-and-software-sharing#loc-sharing-code and ensure that your code is shared in a way that follows best practice and facilitates reproducibility and reuse.

Regarding the code sharing requirements, we confirm that all author-generated code underpinning the findings in our manuscript has been made available without restrictions. We have uploaded all computational scripts and analytical code as supplementary files with our submission, including scripts for differential gene expression analysis, WGCNA analysis, machine learning models, and single-cell RNA-seq analysis.

We sincerely apologize for the discrepancy. We have now corrected and unified the grant information in both the 'Funding Information' and 'Financial Disclosure' sections to ensure complete consistency throughout the manuscript.

“This study was supported by the talent Program of Fushun City of China (No. FSYC202107012).”

We have amended our financial disclosure statement as requested. The complete and correct statement is: "This study was supported by the talent Program of Fushun City of China (No. FSYC202107012). The funders had no role in study design, data collection and analysis, decision to publish, or preparation of the manuscript." We have updated this statement in our cover letter accordingly.

6. We note that your Data Availability Statement is currently as follows: All relevant data are within the manuscript and its Supporting Information files.

Regarding the Data Availability Statement, we would like to clarify that our study is based entirely on publicly available data from the GEO database. These datasets can be freely accessed by any researcher at https://www.ncbi.nlm.nih.gov/geo/. We have included complete information about these data sources, including their accession numbers and detailed descriptions, both in our manuscript's Methods section and in the supplementary files. Additionally, we have uploaded all analysis code to the Supporting Information files, which includes clear documentation of how to access and process these public datasets. The code files contain all necessary scripts to replicate our bioinformatics analyses, including the values behind means, standard deviations, and other measures reported, as well as the scripts used to generate all graphs and figures. Since all raw data are publicly accessible through GEO, and our analysis code is provided in the supplementary files, researchers have everything needed to fully replicate our study findings. Besides, we also have updated our Data Availability Statement in the manuscript to read: “All raw data are available from the corresponding author upon request.”

7. PLOS requires an ORCID iD for the corresponding author in Editorial Manager on papers submitted after December 6th, 2016. Please ensure that you have an ORCID iD and that it is validated in Editorial Manager. To do this, go to ‘Update my Information’ (in the upper left-hand corner of the main menu), and click on the Fetch/Validate link next to the ORCID field. This will take you to the ORCID site and allow you to create a new iD or authenticate a pre-existing iD in Editorial Manager.

We confirm that the corresponding author's ORCID iD has been validated in the Editorial Manager system as required. The ORCID iD is now properly linked and authenticated in our submission.

8. PLOS ONE now requires that authors provide the original uncropped and unadjusted images underlying all blot or gel results reported in a submission’s figures or Supporting Information files. This policy and the journal’s other requirements for blot/gel reporting and figure preparation are described in detail at https://journals.plos.org/plosone/s/figures#loc-blot-and-gel-reporting-requirements and https://journals.plos.org/plosone/s/figures#loc-preparing-figures-from-image-files. When you submit your revised manuscript, please ensure that your figures adhere fully to these guidelines and provide the original underlying images for all blot or gel data reported in your submission. See the following link for instructions on providing the original image data: https://journals.plos.org/plosone/s/figures#loc-original-images-for-blots-and-gels.

We thank the editor for this important reminder regarding blot/gel reporting requirements. We have now prepared and uploaded the original, uncropped, and minimally adjusted images for all Western blot results reported in our manuscript. All original blot images are now available in the Supporting Information file S1_raw_images.pdf. And we have added the description to the cover letter.

9. We notice that your supplementary figures are uploaded with the file type 'Figure'. Please amend the file type to 'Supporting Information'. Please ensure that each Supporting Information file has a legend listed in the manuscript after the references list.

We have now amended the file type of all supplementary figures from 'Figure' to 'Supporting Information' in the submission system. Additionally, we have ensured that each Supporting Information file has a corresponding legend listed in the manuscript after the references list, as required by PLOS ONE guidelines.

We have carefully reviewed all reviewer comments and confirm that no specific previously published works were recommended for citation by the reviewers. Therefore, no additional citations based on reviewer recommendations have been added to the revised manuscript.

Reviewer #1: In the present study authors performed comptehensive bioinformatic and experimental analysis of the role of STAT1 in the heart failure and the effect of dapagliflozin. The results show that heart failure induced by coronary artery ligation is associated with the up-regulation of myocardial STAT1 whereas dapagliflozin reduced its expression. Dapagliflozin reduced the abundance of circulating M1 and increased M2 macrophages in rats with heart failure. DAPA lso reduced the level of inflammatory cytokines and total bile acids. In vitro, STAT1 overexpression reduced viability of H9C2 cardiomyocytes; the effect suppressed by dapagliflozin.

The topic and the results are of interest, however, there are also some important concerns to be addressed.

Dear Reviewer #1,

We sincerely thank you for your thorough and insightful review of our manuscript. Your comprehensive evaluation and constructive suggestions have substantially improved the quality and rigor of our work. We are particularly grateful for your attention to methodological details, statistical considerations, and interpretative nuances that we had not adequately addressed in the original submission.

Your thoughtful comments regarding the generalizability of our findings to the SGLT2 inhibitor class, the importance of sex differences in heart failure research, and the distinction between biomarker identification and mechanistic validation have prompted us to think more critically about the scope and limitations of our study. These insights have not only strengthened our current manuscript but have also deepened our understanding of the research questions and guided our thinking about future experimental directions.

We have carefully addressed each of your comments point-by-point and made corresponding revisions throughout the manuscript. The detailed responses to your specific comments are provided below. We believe these revisions have significantly enhanced the scientific rigor, transparency, and interpretative accuracy of our work.

Once again, we greatly appreciate the time and expertise you devoted to reviewing our manuscript. Your rigorous evaluation has been invaluable in refining our research.

1. Only one SGLT inhibitor was used in the experiments. Could the results be extrapolated to the whole class of drugs or apply only do dapagliflozin?

Thank you for raising this insightful question regarding the generalizability of our findings. We agree that since only dapagliflozin (DAPA) was used in our experimental setup, the direct conclusion that DAPA mitigates heart failure through STAT1 inhibition strictly applies to this specific agent.

However, based on extensive clinical evidence and proposed mechanistic overlap, we believe there is a strong rationale for a potential class effect concerning this pathway. The SGLT2 inhibitor class (including DAPA and Empagliflozin) has consistently demonstrated a class effect in major clinical trials (such as DAPA-HF and EMPEROR trials), significantly reducing heart failure hospitalization and cardiovascular death across different patient populations, irrespective of diabetes status. This suggests that the cardioprotective benefits are driven by mechanisms common to the class. Given that STAT1 is a crucial mediator of inflammation and metabolic stress—both universal pathological features of heart failure targeted by SGLT2 inhibitors—we hypothesize that STAT1 suppression represents a shared, pleiotropic, downstream target for the entire SGLT2 inhibitor class.

Nonetheless, we acknowledge this as a limitation of our current study, and we have added a statement to the Discussion section to address this point and suggest future comparative studies to definitively confirm the proposed class effect: “The robust evidence of a class effect for SGLT2 inhibitors in reducing HF hospitalization and cardiovascular mortality[37, 38] suggests that the fundamental mechanisms, such as STAT1 inhibition and subsequent immune/metabolic modulation, may be shared across the class. Future studies directly comparing the effects of different SGLT2 inhibitors (e.g., Dapagliflozin vs. Empagliflozin) on the STAT1 pathway are warranted to definitively confirm this proposed class effect.”

2. The method of LAD ligation including anesthesia and recovery should be described in more details.

We thank the reviewer for this important suggestion. We have now expanded the Methods section to provide more detailed descriptions of the LAD ligation procedure, including anesthesia protocols and post-operative recovery procedures:

"Myocardial infarction–induced heart failure (HF) was created in the experimental group by ligation of the left anterior descending (LAD) coronary artery. Specifically, rats were anesthetized with isoflurane and fixed in the supine position after hair removal. Electrocardiogram (ECG) was continuously monitored throughout the procedure. After sterile preparation, a thoracotomy was performed along the left sternal border at the 3rd–4th intercostal space using tissue scissors to expose the heart. The left atrial appendage was gently lifted with sterile toothed forceps to expose the aortic root. A surgical suture was passed around the proximal segment of the left anterior descending coronary artery near the superior margin of the heart, and three secure knots were tied to achieve complete ligation. The thoracic wall was then rapidly closed with sutures. Control rats underwent the same thoracotomy procedure without LAD ligation (sham operation). Post-operatively, all rats were placed in a warm recovery chamber and monitored until full consciousness was regained. Analgesics and antibiotics were administered as needed to minimize discomfort and prevent infection."

3. Sex of the animals used in the experiments sho

---

## [Decision Letter · Decision Letter 1]

5 Feb 2026

Dapagliflozin mitigates myocardial inflammation and metabolic stress in heart failure through STAT1 inhibition: evidence from multi-omics analyses and experimental exploration

PONE-D-25-51193R1

Dear Dr. Song,

We’re pleased to inform you that your manuscript has been judged scientifically suitable for publication and will be formally accepted for publication once it meets all outstanding technical requirements.

Kind regards,

Jian Wu, M.D, Ph.D

Academic Editor

PLOS One

Additional Editor Comments (optional):

Reviewers' comments:

Reviewer's Responses to Questions

**Comments to the Author**

Reviewer #1: All comments have been addressed

Reviewer #2: All comments have been addressed

Reviewer #3: (No Response)

2. Is the manuscript technically sound, and do the data support the conclusions?

Reviewer #1: Yes

Reviewer #2: Yes

Reviewer #3: Partly

3. Has the statistical analysis been performed appropriately and rigorously?

Reviewer #1: Yes

Reviewer #2: Yes

Reviewer #3: No

4. Have the authors made all data underlying the findings in their manuscript fully available?

Reviewer #1: Yes

Reviewer #2: Yes

Reviewer #3: No

5. Is the manuscript presented in an intelligible fashion and written in standard English?

Reviewer #1: Yes

Reviewer #2: Yes

Reviewer #3: No

Reviewer #1: The manuscript has been revised according to the reviewers' comments. All concerns raised by the reviewers have been adequately addressed by the authors.

Reviewer #2: no further comments, all issues have been addressed xxxxxxxxxxxxxxxxxxxxxxxxxxxxxxxxxxxxxxxxxxxxxxxxxxxxxxxxxxxx

Reviewer #3: Although the manuscript addresses a potentially relevant topic in papillary thyroid carcinoma, the overall quality of data presentation and methodological rigor is insufficient. Key figures are low-resolution and heavily cropped, uncropped original blots and raw data are not provided, and the Data Availability statement does not comply with PLOS ONE policies. In addition, experimental design and statistical reporting lack essential details, and the mechanistic conclusions are overstated relative to the supporting evidence. Addressing these issues would require substantial reanalysis and additional work beyond a reasonable revision; therefore, I recommend rejection.

**Do you want your identity to be public for this peer review?** For information about this choice, including consent withdrawal, please see our Privacy Policy

Reviewer #1: No

Reviewer #2: No

Reviewer #3: No

---

## [Editor Report · Acceptance letter]

PONE-D-25-51193R1

PLOS One

Dear Dr. Song,

I'm pleased to inform you that your manuscript has been deemed suitable for publication in PLOS One. Congratulations! Your manuscript is now being handed over to our production team.

Kind regards,

on behalf of

Dr. Jian Wu

Academic Editor

PLOS One